# High-pressure synthesis of ultraincompressible hard rhenium nitride pernitride $Re_2(N_2)(N)_2$ stable at ambient conditions

Maxim Bykov [1], Stella Chariton[1], Hongzhan Fei [1], Timofey Fedotenko[2], Georgios Aprilis [2], Alena V. Ponomareva[3], Ferenc Tasnádi[4], Igor A. Abrikosov[4], Benoit Merle [5], Patrick Feldner[5], Sebastian Vogel[6], Wolfgang Schnick[6], Vitali B. Prakapenka[7], Eran Greenberg [7], Michael Hanfland[8], Anna Pakhomova [9], Hanns-Peter Liermann [9], Tomoo Katsura[1], Natalia Dubrovinskaia [2] & Leonid Dubrovinsky [1]

High-pressure synthesis in diamond anvil cells can yield unique compounds with advanced properties, but often they are either unrecoverable at ambient conditions or produced in quantity insufficient for properties characterization. Here we report the synthesis of metallic, ultraincompressible ($K_0 = 428(10)$ GPa), and very hard (nanoindentation hardness 36.7(8) GPa) rhenium nitride pernitride $Re_2(N_2)(N)_2$. Unlike known transition metals pernitrides $Re_2(N_2)(N)_2$ contains both pernitride $N_2^{4-}$ and discrete $N^{3-}$ anions, which explains its exceptional properties. $Re_2(N_2)(N)_2$ can be obtained via a reaction between rhenium and nitrogen in a diamond anvil cell at pressures from 40 to 90 GPa and is recoverable at ambient conditions. We develop a route to scale up its synthesis through a reaction between rhenium and ammonium azide, $NH_4N_3$, in a large-volume press at 33 GPa. Although metallic bonding is typically seen incompatible with intrinsic hardness, $Re_2(N_2)(N)_2$ turned to be at a threshold for superhard materials.

[1] Bayerisches Geoinstitut, University of Bayreuth, Universitätstraβe 30, 95440 Bayreuth, Germany. [2] Material Physics and Technology at Extreme Conditions, Laboratory of Crystallography, University of Bayreuth, Universitätstraβe 30, 95440 Bayreuth, Germany. [3] Materials Modeling and Development Laboratory, National University of Science and Technology 'MISIS', Leninskiy prospekt 4, Moscow, Russia 119049. [4] Department of Physics, Chemistry and Biology (IFM), Linköping University, Campus Valla, Fysikhuset, SE-58183 Linköping, Sweden. [5] Materials Science and Engineering, Institute I, Friedrich-Alexander-Universität Erlangen-Nürnberg (FAU) Martensstraβe. 5, D-91058 Erlangen, Germany. [6] Chair in Inorganic Solid State Chemistry, Department of Chemistry, University of Munich (LMU), Butenandtstraβe 5-13 (D), D-81377 Munich, Germany. [7] Center for Advanced Radiation Sources, University of Chicago, 5640 S. Ellis, Chicago, IL 60637, USA. [8] European Synchrotron Radiation Facility, BP 220, 38043 Grenoble Cedex, France. [9] Photon Science, Deutsches Elektronen-Synchrotron, Notkestraβe 85, 22607 Hamburg, Germany. Correspondence and requests for materials should be addressed to M.B. (email: maks.byk@gmail.com) or to I.A.A. (email: igor.abrikosov@liu.se)

According to the approach formulated by Yeung et al.[1], the design of novel superhard materials should be based on the combination of a metal with high valence electron density with the first-row main-group elements, which form short covalent bonds to prevent dislocations. This conclusion was based on the synthesis of hard borides, such as $OsB_2$[2], $ReB_2$[3–5], $FeB_4$[6], or $WB_4$[7], whose crystal structures possess covalently bonded boron networks. Similar to boron, nitrogen as well can form covalent nitrogen–nitrogen bonds, but there are several factors, which make it difficult to synthesize nitrogen-rich nitrides. The large bond enthalpy of the triply bound $N_2$ molecule (941 kJ·mol$^{-1}$)[8] makes this element generally unreactive. In many reactions the activation barrier for $N_2$ bond breaking requires temperatures, which are higher than the decomposition temperatures of the target phases. $MN_x$ compounds with $x > 1$ are rarely available via direct nitridation reactions or ammonothermal syntheses[9,10]. Therefore, binary $M$-N systems are often limited to interstitial metal-rich nitrides. Usually, they are less compressible and have higher bulk moduli in comparison with pure metals due to the increasing repulsion between metal and nitrogen atoms, whereas their shear moduli are not always much different from those of metals.

Application of pressure is one way to increase the chemical potential of nitrogen and to stabilize nitrogen-rich phases[11]. Several transition metal dinitrides, $PtN_2$[12], $PdN_2$[13], $IrN_2$[14], $OsN_2$[14], $TiN_2$[15], $RhN_2$[16], $RuN_2$[17], $CoN_2$[18], and $FeN_2$[19], containing covalently bound dinitrogen units were recently synthesized in laser-heated diamond anvil cells (LHDACs) via reactions between elemental metal and nitrogen in a pressure range of 40–80 GPa. Although LHDAC is an efficient method to study high-pressure chemical reactions, it is challenging to scale up the synthesis. The search for suitable synthetic strategies, which would enable an appropriate reaction to be realized in a large volume press (LVP) instead of a LHDAC, is an important challenge for high-pressure chemistry and materials sciences. In this study, focusing on the high-pressure synthesis of nitrogen-rich phases in the Re-N system and the development of new synthetic strategies, we resolved this problem for a rhenium nitride $ReN_2$ with unusual crystal chemistry and unique properties.

Direct reactions between rhenium and nitrogen were studied by Friedrich et al.[20], who synthesized two interstitial rhenium nitrides $Re_3N$ at 13 GPa and 1700 K, and $Re_2N$ at 20 GPa and 2000 K. Both compounds have exceptionally large bulk moduli exceeding 400 GPa (as measured upon compression in a non-hydrostatic medium[21]), but only moderate shear moduli as expected for interstitial compounds[22]. Kawamura et al.[23] reported synthesis of $ReN_2$ with $MoS_2$ structure type ($m$-$ReN_2$) in a metathesis reaction between $Li_3N$ and $ReCl_5$ at 7.7 GPa. Subsequently, Wang et al.[21] suggested, based on the first-principle calculations, that $m$-$ReN_2$ is unstable and 'real stoichiometric' $ReN_2$ should have monoclinic $C2/m$ symmetry and transform to the tetragonal $P4/mbm$ phase above 130 GPa. However, this suggestion has not been proven experimentally as yet. Recently Bykov et al. reported a novel inclusion polynitrogen compound

$ReN_8·xN_2$ synthesized from elements at 106 GPa[24], but the region of ~35–100 GPa still remains completely unexplored for the Re-N system.

Here, we report the high-pressure synthesis of an ultra-incompressible metallic hard compound $ReN_2$ via reactions between rhenium and nitrogen or ammonium azide at pressures of 33–86 GPa. The usage of a solid nitrogen precursor $NH_4N_3$ allows to scale up the synthesis of $ReN_2$ in the large volume press.

## Results

**Synthesis of $Re_2(N_2)(N)_2$ in a laser-heated diamond anvil cell.** We have studied chemical reactions between Re and nitrogen and other reagents, such as sodium azide $NaN_3$ and ammonium azide $NH_4N_3$, in LHDACs in a range of 29–86 GPa at temperatures of 2000–2500 K (Table 1, Experiments #1 through #6). The reactions products typically contained numerous single-crystalline grains of several rhenium nitride phases (Table 1), which were identified using synchrotron single-crystal X-ray diffraction (Supplementary Note 1, Supplementary Fig. 1, Supplementary Tables 1–3).

A direct reaction between Re and $N_2$ (Table 1) resulted in the synthesis of three rhenium nitrides $ReN_2$, $Re_2N$, and $ReN_{0.6}$, two of which ($ReN_2$ and $ReN_{0.6}$) have never been observed before. The third phase identified in these experiments, $Re_2N$ ($P6_3/mmc$), has previously been reported[20]. After a stepwise decompression of the sample obtained in Experiment #1 down to the ambient pressure, all of the three phases ($ReN_2$, $ReN_{0.6}$, $Re_2N$) were found to remain intact even after being exposed to atmospheric oxygen and moisture for several months. Crystal structure analysis of $ReN_{0.6}$ showed that it has a defect WC structure type (space group $P\bar{6}m2$) (for details on $ReN_{0.6}$ see Supplementary Note 2, Supplementary Figs. 2 and 3, Supplementary Table 4).

Analysis of the crystal structure of $ReN_2$ revealed its unusual crystal-chemistry. Figure 1 shows the crystal structure of $ReN_2$, which is built of distorted $ReN_7$ capped trigonal prisms (Fig. 1d) and contains both N–N units (dumbbells) (Fig. 1f) and discrete N atoms (N2) (Fig. 1e) in an atomic ratio 1:1. The N1–N1 dumbbells are located in a trigonal antiprism formed by Re atoms (Fig. 1f), while discrete N2 atoms have a tetrahedral coordination by Re (Fig. 1e). The N1–N1 bond length ($d_{N1-N1} = 1.412(16)$ Å at ambient conditions) suggests that the $N_2$ unit should be considered as a pernitride anion $N_2^{4-}$. Therefore, $ReN_2$ is a rhenium nitride pernitride and its crystal-chemical formula is $Re^{+V}_2[N^{-II}_2][N^{-III}]_2$. In the following discussion we interchangeably use both empirical formula $ReN_2$ and crystal-chemical formula for this compound.

**Compressibility of $Re_2(N_2)(N)_2$.** The compressibility of $ReN_2$ was measured on the sample #2 (Table 1), which was synthesized at 49 GPa, then decompressed down to ambient conditions, and re-loaded into another DAC with a neon pressure-transmitting medium, which provides much better hydrostaticity of the sample environment than nitrogen[25]. The sample was first characterized using single-crystal XRD at ambient conditions. On compression,

### Table 1 Summary of syntheses

| Experiment | Technique | Reagents | Pressure (GPa) | Temperature (K) | Products |
|---|---|---|---|---|---|
| 1 | LHDAC | Re + $N_2$ | 42 | 2200 (300) | $ReN_2$ + $Re_2N$ + $ReN_{0.6}$ |
| 2 | LHDAC | Re + $N_2$ | 49 | 2200 (300) | $ReN_2$ + $Re_2N$ |
| 3 | LHDAC | Re + $N_2$ | 71 | 2500 (300) | $ReN_2$ + $Re_2N$ |
| 4 | LHDAC | Re + $N_2$ | 86 | 2400 (300) | $ReN_2$ + $Re_2N$ |
| 5 | LHDAC | Re + $NaN_3$ | 29 | 2000 (300) | $NaReN_2$ + $Re_2N$ |
| 6 | LHDAC | Re + $NH_4N_3$ | 43 | 2200 (300) | $ReN_2$ + $ReN_{0.6}$ + $Re_2N$ |
| 7 | LVP | Re + $NH_4N_3$ | 33 | 2273 (100) | $ReN_2$ + $Re_2N$ |

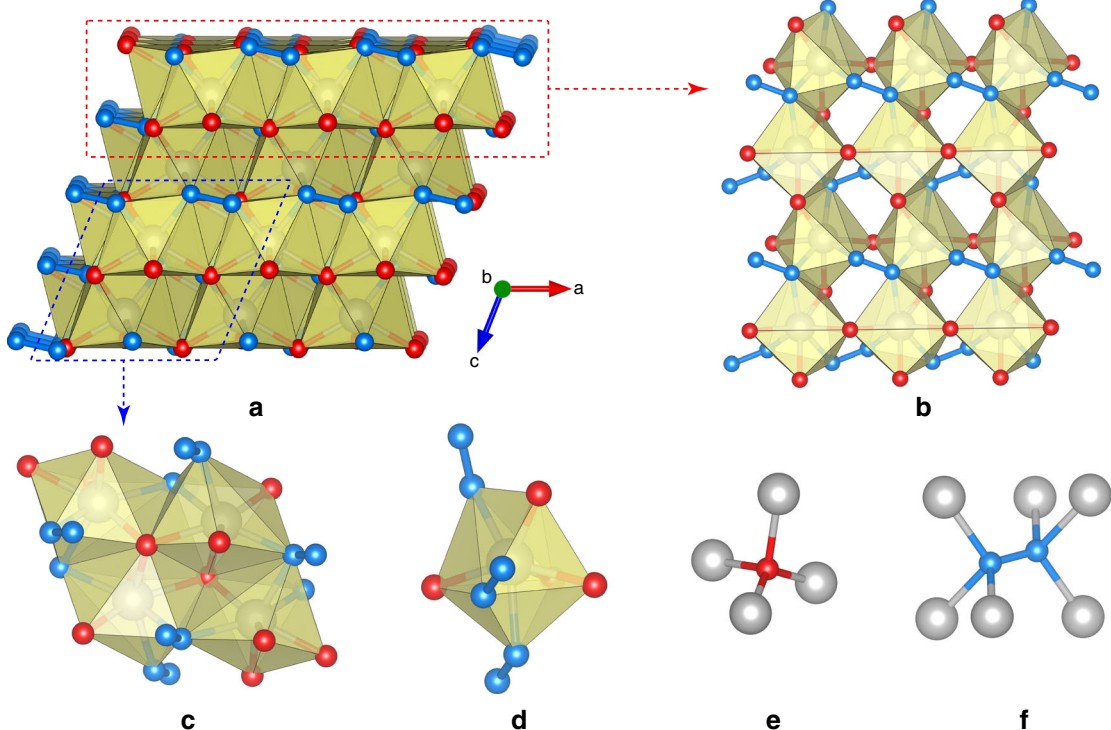

**Fig. 1** Fragments of the crystal structure of $Re_2(N_2)(N)_2$ at ambient conditions. Re atoms–gray, N1 atoms–blue, N2 atoms–red. $ReN_2$ crystallizes in the space group $P2_1/c$ (No. 14) with $a = 3.6254(17)$, $b = 6.407(7)$, $c = 4.948(3)$ Å, $\beta = 111.48(6)°$. Rhenium and nitrogen atoms occupy Wyckoff Positions 4e: Re [0.35490(11), 0.34041(8), 0.19965(8)], N1 [0.194(2), 0.038(2), 0.311(19)], N2 [0.259(3), 0.6381(18), 0.024(2)]. Full crystallographic information is given in the supplementary crystallographic information file and in the Supplementary Tables 2 and 3. **a** The projection of the crystal structure along the b-axis. **b**, **c** Fragments of the crystal structure of $ReN_2$ showing how $ReN_7$ polyhedra are connected with each other. **d** Separate $ReN_7$ coordination polyhedron. **e** Coordination of N2 atoms. **f** Coordination of N1–N1 dumbbells

the lattice parameters were extracted from the powder XRD data (Fig. 2a, b; Supplementary Note 3, Supplementary Figs. 4-6; Supplementary Table 5). The pressure-volume dependence was described using the third-order Birch–Murnaghan equation of state[26] with the following fit parameters: $V_0 = 107.21(4)$ Å$^3$, $K_0 = 428(10)$ GPa, $K' = 1.6(5)$. Figure 2c shows a plot of correlated values of $K_0$ and $K'$ to different confidence levels. The bulk modulus $K_0$ lies within the range of 410–447 GPa at the 99.73% confidence level. Thus, $K_0$ of $ReN_2$ is larger than that of any compound in the Re-N system and is comparable to that of diamond ($K_0 = 440$ GPa) and $IrN_2$ ($K_0 = 428(12)$ GPa)[14]. Among very incompressible pernitrides of transition metals, $ReN_2$ is the only compound, in which the metal atom has oxidation state (+V) higher than (+IV). The enhancement of the bulk modulus of $ReN_2$ in comparison to $OsN_2$, $PtN_2$, and $TiN_2$ is therefore in agreement with the general trend, that the bulk modulus of a compound increases with an increase of the product of formal charges of anions and cations[19,27].

**Search for solid nitrogen precursor**. More detail characterization of physical properties of $ReN_2$, such as hardness, electrical conductivity, etc. require a sample to be at least a few tens of microns in size that is much larger than can be synthesized in a LHDAC. The large volume press technique enables the synthesis of such a sample, but precludes from using $N_2$ as a reagent. First, the amount of nitrogen, which can be sealed in a capsule along with Re, is insufficient for the desired reaction yield; second, unavoidable deformation of the capsule upon compression may potentially lead to the loss of nitrogen. Therefore, a solid source of nitrogen had to be found and we tested sodium and ammonium azides, $NaN_3$ and $NH_4N_3$, as potential precursors in LHDACs

(Experiments #5, #6, Table 1) (for a discussion regarding the choice of the solid reagents see Supplementary Note 4). The experiment with $NaN_3$ (Experiment #5) did not result in the synthesis of $ReN_2$. The major product of the reaction was $NaReN_2$ (Supplementary Fig. 7), whose lattice parameters turned out to be very close to those reported for m-$ReN_2$ by Kawamura et al.[23], that might suggest that the material described in ref.[23] as rhenium nitride indeed could be a different compound (for a related discussion see Supplementary Note 4). The experiment in LHDAC with $NH_4N_3$ as a source of nitrogen resulted in the synthesis of $ReN_2$ among other products (Experiment #6, Table 1).

**Synthesis of $Re_2(N_2)(N)_2$ in a large volume press**. Based on results of this experiment in DAC, we explored a possibility to scale up the synthesis of $ReN_2$ in a multianvil LVP at 33 GPa and 2273 K via a reaction between rhenium and ammonium azide (Experiment #7, Supplementary Note 5, Supplementary Fig. 8). The product of the reaction was a mixture of $Re_2N$ and $ReN_2$. Each phase was separated (Supplementary Fig. 9) and characterized using single-crystal X-ray diffraction. A phase-pure polycrystalline sample of $ReN_2$ ($70 \times 60 \times 50$ μm$^3$), which was synthesized in the LVP, was used for nanoindentation hardness and electrical resistance measurements (Supplementary Fig. 10). Nanoindentation was performed using a nanoindenter equipped with Berkovich diamond tip and featuring continuous stiffness measurement capabilities. The average hardness and Young's modulus measured between 200 and 400 nm depths are 36.7(8) GPa and 493(14) GPa, respectively (Fig. 2e, Table 2). The hardness approaching 40 GPa, a threshold for superhard materials, and extreme stiffness comparable with diamond makes

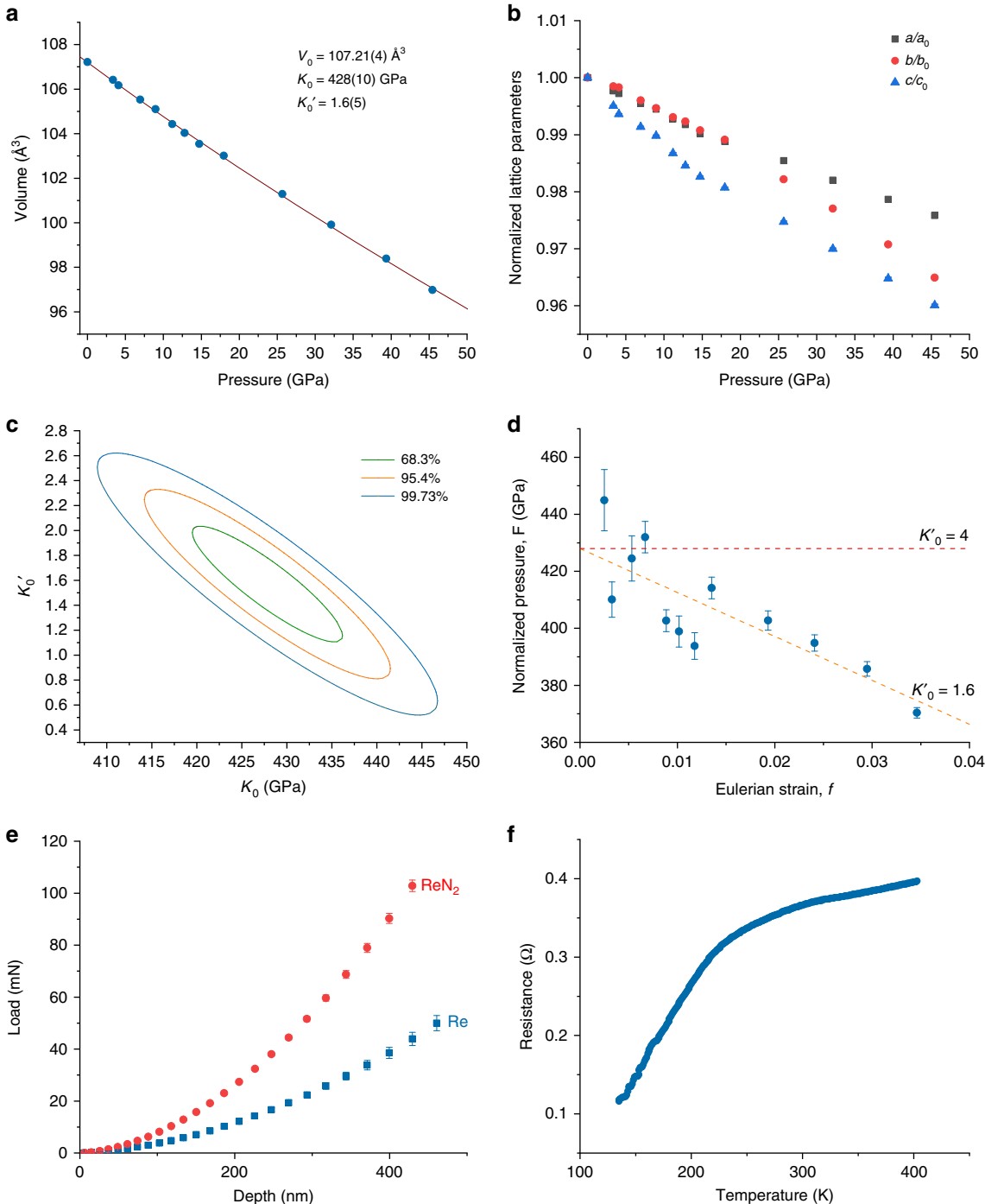

**Fig. 2** Physical propeties of $Re_2(N_2)(N)_2$. **a** Pressure-dependence of the unit-cell volume of $ReN_2$. The solid line shows the fit of the third- order Birch–Murnaghan EoS to the experimental data. **b** Normalized lattice parameters of $ReN_2$. Black squares $a/a_0$, red circles–$b/b_0$, blue triangles–$c/c_0$, where $a_0$, $b_0$, $c_0$ are the lattice parameters at ambient conditions. **c** Plot of the correlated values of $K_0$ and $K_0'$ to different confidence levels of 68.3%, 95.4%, and 99.73%, respectively. **d** An F-f plot based on Birch–Murnaghan EoS. Dashed lines indicate expected behavior of data points for certain $K_0'$ values. Uncertainties were calculated by propagation of experimental uncertainties in the unit cell volume. **e** Averaged indentation load-displacement data for Re (blue squares) and for $ReN_2$ (red circles). The error bars correspond to the standard deviation between 16 single measurements at different locations. **f** Temperature dependence of the electrical resistance of the $ReN_2$ sample at ambient pressure. If error bars are not shown, they are smaller than the symbol size

mechanical properties of $ReN_2$ exceptional in the row of metal nitrides. Due to the directional N–N bonding, the hardness of $ReN_2$ is higher than that of known interstitial transition metal nitrides (δ-NbN –20 GPa, HfN –19.5 GPa, ZrN –17.4 GPa[28], CrN −17 GPa[1] etc.). Most transition metal pernitrides $MN_2$ that are metastable at ambient conditions are expected to be very hard

compounds too, however they were never obtained in a quantity sufficient for the hardness measurements[29,30].

The electrical resistance of $ReN_2$ as a function of temperature was measured at ambient pressure on a sample with the dimensions of about $70 \times 60 \times 50\,\mu m^3$. The results of the measurements in the range of 150 K to 400 K are shown in

| Table 2 Mean hardness and Young's modulus of ReN₂ measured by nanoindentation in the 200–400 nm depth range | | |
| --- | --- | --- |
| **Material** | **Hardness (GPa)** | **Young's modulus (GPa)** |
| ReN$_2$ | 36.7 (8) | 493 (14) |
| Re | 10.9 (6) | 424 (12) |
| Note: The error estimate corresponds to the standard deviation between 16 different locations | | |

Fig. 2f. Electrical resistivity of metals increases with temperature and this is the case for ReN$_2$. The shape of the resistance–temperature curve (Fig. 2f) is reproducible as confirmed in a number of independent measurements on the same sample with re-glued electrical contacts.

**Theoretical calculations**. To confirm the experimentally observed peculiarities of ReN$_2$ and to gain deeper insights into the mechanical and electronic properties of this compound, we performed theoretical calculations based on the density functional theory. First, we considered the crystal structure of ReN$_2$. We carried out the full structure optimization for the compound at ambient pressure and found that calculations and experiment are in very close agreement (Supplementary Table 2). Calculated elastic constants for ReN$_2$ (Table 3) fulfill the mechanical stability conditions[31], and calculated phonon dispersion relations (Fig. 3a) show only real frequencies confirming its dynamic stability. Theoretically calculated N1–N1 vibrational frequency form a localized band giving rise to a peak of the phonon density of states at ~1031 cm$^{-1}$. This vibrational behavior is similar to other pernitrides[12,15,32]. The metallic nature of the material confirmed by our calculations of the electronic density of states (DOS) (Fig. 3c). Calculated vibrational and electronic properties of N1–N1 unit confirm that it is a pernitride anion $N_2^{4-}$. On the contrary, electronic and vibrational properties of N2 atoms (Fig. 3a, c) are quite distinct from those of N1, providing strong support to the experimental observation of the crystal chemistry of ReN$_2$, which is unique for transition metals pernitrides.

**Discussion**

The unique chemistry of the compound is essential for understanding its superior mechanical properties. The bulk modulus, calculated theoretically using Voigt-Reuss-Hill approximation (413.5 GPa)[33] is in a good agreement with the experiment (428 (10) GPa), confirming that ReN$_2$ can be characterized as highly incompressible material. At the same time, the value of the Poisson coefficient is close to 0.25, and relatively high ratio between share and bulk elastic moduli indicates substantial degree of covalence in ReN$_2$ chemical bonding. A direct calculation of the charge density map (Fig. 3b) confirms the expectation. One sees a formation of covalent bonds between two N1 atoms. It is of single bond character with very high degree of electron localization (Fig. 3d) typical for a pernitride anion $N_2^{4-}$ in other transition metal pernitrides[34], and incompressible $N_2^{4-}$ is supposed to contribute to very low compressibility of the materials. The covalent bond between Re and N2 atoms is formed by substantially less localized electrons (compare Fig. 3b and Fig. 3d). Indeed, measured temperature dependence of the electrical resistance (Fig. 2f) and the estimated resistivity (~$4 \times 10^{-6}$ Ω·m–$16 \times 10^{-6}$ Ω·m in a temperature range 150–400 K) are in agreement with the theoretical conclusion and the description of ReN$_2$ as a metal. The formation of the covalent bonds between Re and N2 atoms indicates strong hybridization between the electronic states of the atoms. The calculated electronic DOS (Fig. 3c)

shows the presence of the pseudogap between occupied, predominantly bonding states of Re and unoccupied non-bonding and anti-bonding states. According to Jhi et al.[35], such features optimize electronic contribution to hardness enhancement in transition-metal carbonitrides, which can also explain very high hardness of ReN$_2$. Thus, the formation of strong covalent bond between Re and N2 atoms, a unique feature of the material synthesized in this work in comparison with known transition metal pernitrides, appears to be essential for its spectacular mechanical and electronic properties. To summarize, in the present work we have synthesized a transition metal nitride ReN$_2$ ($Re^{+V}_2(N^{-II}_2)(N^{-III})_2$) with the unique crystal structure and outstanding properties. The structure with Re atoms in the high oxidation state +V features both discrete nitride and pernitride ions. A combination of the high electron density of the transition metal with interstitial nitride anions and covalently bound pernitride units makes this compound ultraincompressible and extremely hard at the same time. The developed method for scaling up the synthesis of ReN$_2$ in a LVP using ammonium azide as a nitrogen precursor may be applied for producing nitrides of other transition metals. We demonstrated the complete route for materials development from screening experiments in diamond anvil cells to the synthesis of samples large enough for physical property measurements. It is not only our results per se that are important, but the fact that the development and synthesis of the new material was realized contrary to the established concepts and should encourage further theoretical and experimental works in the field.

**Methods**

**Synthesis of Re-N phases in laser-heated diamond anvil cells**. In all synthesis experiments a rhenium powder (Sigma Aldrich, 99.995%) was loaded into the sample chamber of a BX90 diamond anvil cell (Boehler–Almax anvils, 250-μm size). In four experiments the chamber was filled with nitrogen at 1.5 kbar that served as a pressure-transmitting medium and as a reagent. In two experiments, the chamber was filled either with ammonium azide NH$_4$N$_3$ or with sodium azide NaN$_3$. Pressure was determined using the equation of state of rhenium[36–38]. The compressed sample was heated using the double-sided laser-heating system installed at the Bayerisches Geoinstitut (BGI), University of Bayreuth, Germany. Successful syntheses were performed at 40, 42, 49, 71, and 86 GPa at temperatures of 2200–2500 K (Table 1).

**Synthesis of Re-N phases in the large-volume press**. High-pressure synthesis was performed using a Kawai-type multi-anvil apparatus IRIS15, installed at the BGI[39]. The NH$_4$N$_3$ sample (0.5 mm thickness, 0.8 mm in diameter) was sandwiched between two layers of Re powder (0.1 mm thick, 0.8 mm in diameter) and between two tubes of dense alumina in a Re capsule, which also acted as a heater. The capsule was placed in a 5 wt% Cr$_2$O$_3$-doped MgO octahedron with a 5.7 mm edge that was used as the pressure medium. The assembly scheme is given in the Supplementary Fig. 8. Eight tungsten carbide cubes with 1.5 mm truncation edge lengths were used to generate high pressures. The assembly was pressurized at ambient temperature to 33 GPa, following the calibration given by Ishii et al.[39] and then heated to ~2273(100) K within 5 min and immediately quenched after the target temperature was reached. The assembly was then decompressed during 16 h.

**Synthesis of NH$_4$N$_3$**. Ammonium azide, NH$_4$N$_3$ was obtained by the metathesis reaction between NH$_4$NO$_3$ (2.666 g, 33 mmol, Sigma-Aldrich, 99.0%) and NaN$_3$ (2.165 g, 33 mmol, Acros Organics, Geel, Belgium, 99%) in a Schlenk tube. By heating from room temperature to 170 °C in a glass oven and annealing for 7.5 h at 170 °C and then for 12 h at 185 °C, NH$_4$N$_3$ precipitated at the cold end of the tube separated from NaNO$_3$, which remained at the hot end during the reaction[40].

**Compressibility measurements**. For the compressibility measurements the sample synthesized at 49 GPa and 2200 K (Experiment #2) was quenched down to ambient pressure and re-loaded into another diamond anvil cell. The sample chamber was then filled with Ne that served as a pressure-transmitting medium. A powder of gold (Sigma Aldrich, 99.99%) was placed into the sample chamber along with the sample and used as a pressure standard[41]. The sample was then compressed up to ~45 GPa in 13 steps. At each pressure point we have collected powder X-ray diffraction data.

**Table 3 Calculated elastic properties of ReN₂**

| $C_{11}$ | $C_{12}$ | $C_{13}$ | $C_{15}$ | $C_{22}$ | $C_{23}$ | $C_{25}$ | $C_{33}$ | $C_{35}$ |
|---|---|---|---|---|---|---|---|---|
| 869.51 | 230.73 | 261.47 | 51.01 | 748.93 | 251.83 | 26.67 | 648.06 | 16.61 |
| $C_{44}$ | $C_{46}$ | $C_{55}$ | $C_{66}$ | $B$ | $G$ | $E$ | $\nu$ | |
| 257.43 | 35.91 | 299.94 | 266.34 | 413.5 | 262 | 650 | 0.24 | |

Note: Elastic constants $C_{ij}$ (GPa), bulk modulus $B$ (GPa), shear modulus $G$ (GPa), Young's modulus $E$ (GPa), and Poisson's ratio ($\nu$)

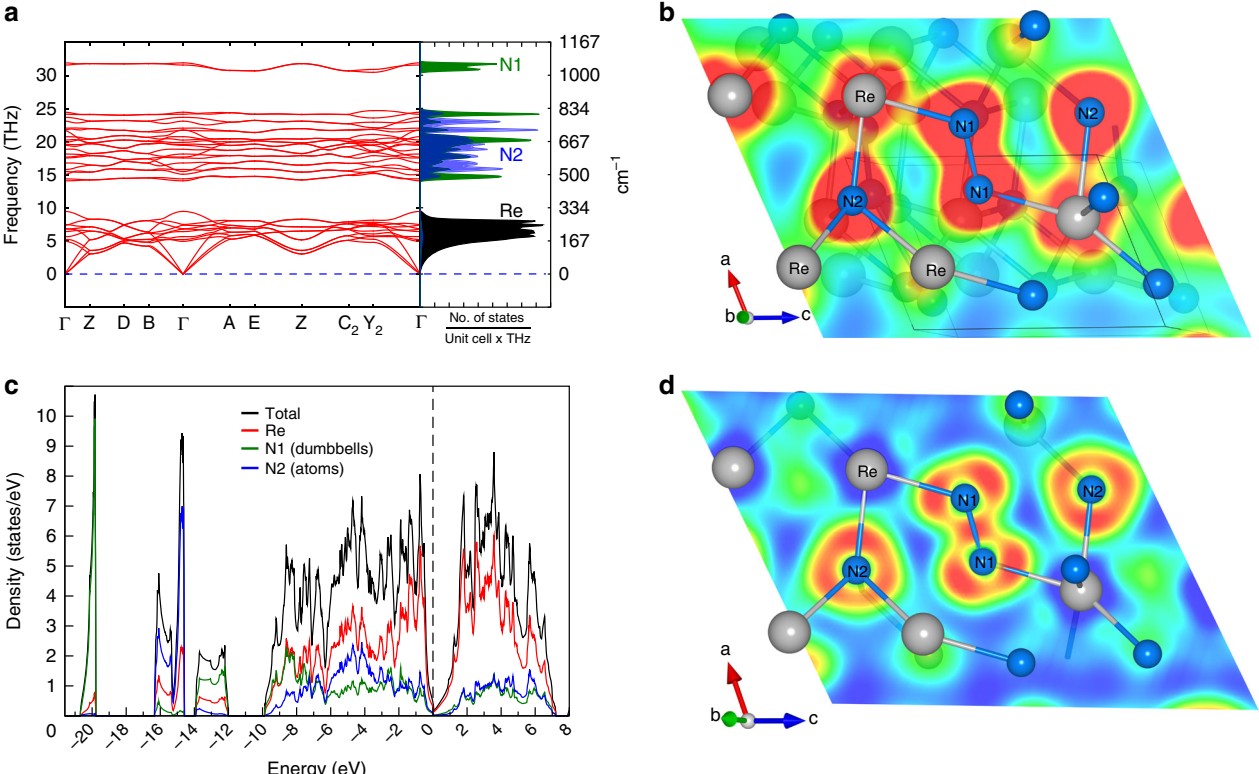

**Fig. 3** Phonon and electronic structure calculations for ReN₂. Calculated phonon dispersion relations (**a**), charge density map (**b**), densities of states (**c**), and electron localization function (**d**) for ReN₂ at ambient conditions

**Synchrotron X-ray diffraction**. High-pressure single-crystal and powder synchrotron X-ray diffraction studies of the reaction products were performed at the beamlines P02.2 (DESY, Hamburg, Germany)[42], ID15B (ESRF, Grenoble, France), and 13-IDD GSECARS beamline (APS, Argonne, USA). The following beamline setups were used. P02.2: $\lambda = 0.29$ Å, beam size ~2 × 2 μm², Perkin Elmer XRD 1621 detector; ID15B: $\lambda = 0.41$, beam size ~10 × 10 μm², Mar555 flat panel detector; GSECARS: $\lambda = 0.2952$ Å, beam size ~3 × 3 μm², Pilatus CdTe 1 M detector. For the single-crystal XRD measurements samples were rotated around a vertical ω-axis in a range ±38°. The diffraction images were collected with an angular step $\Delta\omega = 0.5°$ and an exposure time of 1 s/frame. For analysis of the single-crystal diffraction data (indexing, data integration, frame scaling and absorption correction) we used the *CrysAlis^Pro* software package. To calibrate an instrumental model in the *CrysAlis^Pro* software, i.e., the sample-to-detector distance, detector's origin, offsets of goniometer angles, and rotation of both X-ray beam and the detector around the instrument axis, we used a single crystal of orthoenstatite (($Mg_{1.93}Fe_{0.06}$)($Si_{1.93}$, $Al_{0.06}$)$O_6$, *Pbca* space group, $a = 8.8117(2)$, $b = 5.18320(10)$, and $c = 18.2391(3)$ Å). The same calibration crystal was used at all the beamlines.

Powder diffraction measurements were performed either without sample rotation (still images) or upon continuous rotation in the range ±20°ω. The images were integrated to powder patterns with Dioptas software[43]. Le-Bail fits of the diffraction patterns were performed with the TOPAS6 software.

**In-house X-ray diffraction**. Ambient-pressure single-crystal XRD datasets were collected with a high-brilliance Rigaku diffractometer (Ag Kα radiation) equipped with Osmic focusing X-ray optics and Bruker Apex CCD detector in the BGI.

**Structure solution and refinement**. The structure was solved with the ShelXT structure solution program[44] using intrinsic phasing and refined with the Jana 2006 program[45]. CSD-1897795 contains the supplementary crystallographic data for this paper. These data can be obtained free of charge from FIZ Karlsruhe via www.ccdc.cam.ac.uk/structures.

**Nanoindentation**. Nanoindentation was performed using a Nanoindenter G200 platform (KLA-Tencor, Milpitas, CA, USA), equipped with a Berkovich diamond tip (Synton MDP, Nidau, Switzerland) and featuring the continuous stiffness based method (CSM)[46]. Each sample was indented at 16 different locations separated by a distance of at least 10 μm, so that their plastic zones did not overlap. For each measurement, loading was performed at a constant strain-rate of 0.025 s⁻¹ up to a maximal indentation depth of at least 400 nm. A 2 nm large oscillation superimposed on the loading signal allowed continuously measuring the contact stiffness. The acquired data were evaluated using the Oliver–Pharr method[47,48]. To this purpose, the diamond punch geometry was calibrated from 1000 nm deep references measurements in fused silica and the machine frame stiffness value was refined so as to obtain a constant ratio between stiffness squared and load during indentation of the samples. The conversion of the reduced moduli to a uniaxial Young's moduli was performed assuming a Poisson's ratios of 0.24 and 0.29, respectively for ReN₂ and Re[49].

**Temperature-dependent resistance measurements**. The resistance of the sample was measured by four-probe method passing a constant DC 90 mA current through the sample and measuring both current and voltage drop across the sample. Temperature was measured using the S-type thermocouple.

**Theoretical calculations**. The ab initio electronic structure calculations of $ReN_2$ (12 atoms), ReN (2 atoms), and $ReN_x$ ($2 \times 3 \times 2$ supercell) were performed using the all electron projector-augmented-wave (PAW) method[50] as implemented in the VASP code[51–53]. Among the tested exchange-correlation potentials (PBE[54], PBE-sol[55], AM05[56]) the PBEsol approximation has resulted into the best agreement between the derived experimental and theoretical equation of state. Convergence has been obtained with 700 eV energy cutoff for the plane wave basis and a ($18 \times 10 \times 14$) Monkhorst–Pack $k$-points[57] type sampling of the Brillouin zone for $ReN_2$. Gaussian smearing technique was chosen with smearing of 0.05 eV. The convergence criterion for the electronic subsystem has been chosen to be equal to $10^{-4}$ eV for two subsequent iterations, and the ionic relaxation loop within the conjugated gradient method was stopped when forces became of the order of $10^{-3}$ eV/Å. The elastic tensor $C_{ij}$ has been calculated from the total energy applying (+/−) 1% and 2% lattice distortions. The Born mechanical stability conditions have been verified using the elastic constants. The phonon calculations have been performed within quasiharmonic approximation at temperature $T = 0$ K using the finite displacement approach implemented into PHONOPY software[58]. Converged phonon dispersion relations have been achieved using a ($3 \times 3 \times 3$) supercell with 324 atoms and ($5 \times 5 \times 5$) Monkhorst–Pack $k$-point sampling.

## Data availability

The data that support the findings of this study are available from the corresponding author upon reasonable request. CSD-1897795 contains the supplementary crystallographic data for this paper. These data can be obtained free of charge from FIZ Karlsruhe via www.ccdc.cam.ac.uk/structures. The source data underlying Fig. 2a–f, and Supplementary Figs. 2 a, b are provided as a Source Data file. Single-crystal X-ray diffraction dataset for rhenium nitrides (experiment #2) at 3.5 GPa has been deposited to Figshare (https://figshare.com/) with the accession link https://doi.org/10.6084/m9.figshare.8081582.

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

## Acknowledgements

N.D. and L.D. thank the Deutsche Forschungsgemeinschaft (DFG projects DU 954-11/1 and DU 393-10/1) and the Federal Ministry of Education and Research, Germany (BMBF, grant no. 5K16WC1) for financial support. S.V. and W.S. are grateful for a PhD fellowship granted by the Fonds der Chemischen Industrie (FCI), Germany. B.M. thanks the Deutsche Forschungsgemeinschaft (DFG project ME 4368/7-1) for financial support. Parts of this research were carried out at the Extreme Conditions Beamline (P02.2) at DESY, a member of Helmholtz Association (HGF). Portions of this work was performed at GeoSoilEnviroCARS (The University of Chicago, Sector 13), Advanced Photon Source (APS), Argonne National Laboratory. GeoSoilEnviroCARS is supported by the National Science Foundation-Earth Sciences (EAR-1634415) and Department of Energy-GeoSciences (DE-FG02-94ER14466). This research used resources of the Advanced Photon Source, a U.S. Department of Energy (DOE) Office of Science User Facility operated for the DOE Office of Science by Argonne National Laboratory under Contract No. DE-AC02-06CH11357, as well as resources from the Center for Nanoanalysis and Electron Microscopy (CENEM) at Friedrich-Alexander University Erlangen-Nürnberg. Several high-pressure diffraction experiments were performed on beamline ID15B at the European Synchrotron Radiation Facility (ESRF), Grenoble, France. We thank Sven Linhardt, Gerald Bauer, Dorothea Wiesner, and Alexander Kurnosov for the help with electrical resistance measurements, SEM measurements and gas-loading. Theoretical analysis of chemical bonding was supported by the Russian Science Foundation (Project No. 18-12-00492). Theoretical calculations of structural properties were supported by the Ministry of Science and High Education of the Russian Federation in the framework of Increase Competitiveness Program of NUST "MISIS" (No. K2-2019-001) implemented by a governmental decree dated 16 March 2013, No 211. Financial support from the Swedish Research Council (VR) through Grant No. 2015-04391, the Swedish Government Strategic Research Area in Materials Science on Functional Materials at Linköping University (Faculty Grant SFO-Mat-LiU No. 2009-00971), and the VINN Excellence Center Functional Nanoscale Materials (FunMat-2) Grant 2016–05156 is gratefully acknowledged. This research is also supported by the European Research Council (ERC) under Horizon 2020 research and innovation program (grant agreement No. 787527).

## Author contributions

M.B., L.D., and N.D. designed the research, M.B., L.D., N.D., I.A.A. wrote the paper, M.B., L.D., S.C, T.F., G.A., V.B.P., E.G., M.H., and A.P. H.-P.L. performed X-ray diffraction experiments, M.B. analyzed the X-ray diffraction data, F.T, A.V.P., I.A. performed theoretical calculations, H.F., M.B., T.K., S.V., W.S. performed synthesis in the large volume press and the synthesis of precursors. B.M. and P.F. performed nanoindentation measurements. L.D. performed electrical resistance measurements. All authors contributed to the discussion of the results.

## Additional information

**Competing interests:** The authors declare no competing interests.

