## [Peer Review File · Nature Communications]

Reviewers' comments:

Reviewer #1 (Remarks to the Author):

"The manuscript by Bykov, et al. describes the preparation and characterization of $\text{Re}_2(\text{N}_2)(\text{N})_2$. This compound was first prepared on a small scale in a diamond anvil cell from rhenium metal and nitrogen gas. To prepare a sample on a bulk scale for mechanical properties characterization, the authors needed a new nitrogen source/precursor, NH_4N_3 , which they prepared from a metathesis reaction between sodium azide and ammonium nitrate. The ammonium azide was then combined with rhenium metal in a multi-anvil hot press. The resulting polycrystalline sample was then characterized via X-ray diffraction, and mechanical properties via nanoindentation (to yield a hardness of ~ 36 GPa) and compression under hydrostatic conditions (to yield a bulk modulus of 428 GPa). This paper should be accepted by virtue of their novel solid-state chemistry and mechanical properties, but the following minor points need to be clarified prior to publication.

1) Could the work here be generalized to other late transition metals? E.g. could tungsten nitrides be analogously prepared with the same NH_4N_3 precursor?

2) On page 6, line 138, the authors argue that there is increased ionicity in Re-N bonding, and this may be why ReN_{-2} has a higher bulk modulus when compared to OsN_2 , PtN_2 , and TiN_2 . I do not think ionicity is the appropriate descriptor as ionic character is usually a reflection of the difference in electronegativity, and one would expect that TiN_2 would be more ionic than ReN_2 as titanium (1.54) is less electronegative / more electropositive than rhenium (1.9). To reflect the unequal distribution of electron density, the authors should consider using the descriptor "polarization" instead of ionicity. Ionicity also conflicts with the author's localization functions (see point #4 below).

3) On page 6, lines 142-143, the authors argue that the increase in oxidation state results in an increase in the bulk modulus, as shown in their previous work on FeN_x . However, that is a bit counterintuitive, as lower oxidation states should have more electron density around the metal center, and thus should have a higher bulk modulus, not the other way around. From the author's previous work on FeN_x , one can rationalize the increase in bulk modulus comes from more electron density from more nitrogen, rather than an increase in oxidation state. Can the authors clarify their argument?

4) On page 9, line 229, the authors argue that the "formation of strong covalent bond between Re and N_2 atoms," and this is confirmed with the author's electron localization functions. I agree with this assessment, and this leads me to think that the bonding is not ionic, but rather polar covalent.

5) Can the authors comment on the stability of ReN_2 versus water and air in their main text?

6) Can the authors comment on how the measured density of their polycrystalline compacts compares to the theoretical crystal density in their main text?

7) The shape and size of the grains are extrinsic effects that can affect the nano-indentation hardness. Do the authors see any coarsening or texturing of the polycrystalline grains? The authors

should include an SEM image of their sample. The synthesis of ReN₂ from ammonium azide should release hydrogen gas, do the authors see any porosity from trapped gasses?

Reviewer #2 (Remarks to the Author):

Report on paper

High-pressure synthesis of 1 ultraincompressible hard rhenium nitride pernitride Re₂(N₂)(N)₂ stable at ambient conditions

By Maxim Bykov et al.

The paper describes the high-pressure high-temperature synthesis of a new compound ReN₂ obtained in the search for new hard materials. Its bulk modulus K₀ of 428(10) GPa is at the threshold of superhard materials. Authors determined the structure of ReN₂, characterized its properties in details and scaled up the process of its synthesis from the DAC to a large volume press. Authors also discuss the role of structural features in the nitrogen-rich compounds.

This is a thorough, quite complex and demanding study and it adds a new representative to the group of metal nitrides and to the search for superhard materials. I have checked the crystallographic data and they are consistent.

On the other hand, this continuous search has been continued for some time, as apparent from the whole series of MN_x compounds, and one may ask, what is the real progress in the chemistry and practical aspects of the production of new superhard materials. ReN₂ is not the best, cheapest or technologically easiest to obtain material and for the practical and technological reasons this compound is not particularly exciting. It is interesting for the structural reasons, as a new member of the series. However I doubt if it would attract strong attention expected for Nature Comm articles.

The paper contains lots of data measured and calculated by several methods, but it is more a report of these properties than sublimed analysis explaining general science. The style of the paper could be improved, too; there are also some mistakes (there are no crystallographic sites 4e, but Wyckoff sites e, or general sites in this case, and 4 is the number of positions) and typos, like in line 61/page 2: binary systems M-N systems ...

Reviewer #3 (Remarks to the Author):

This report by Bykov *et al.* titled “High-pressure synthesis of ultraincompressible hard rhenium nitride pernitride $\text{Re}_2(\text{N}_2)_2$ stable at ambient conditions” provides evidence, both experimental and computational, for the formation of a new form of rhenium nitride, namely, $\text{Re}_2(\text{N}_2)_2$. This nitride pernitride compound is shown to be presenting ultra-low compressibility and high hardness. This discovery is unusual as it challenges the current understanding that metallic binding does not translate into high hardness in a MX_2 compound (where “M” represents a metallic element). In order to document their findings, the authors have carried out numerous rhenium nitride synthesis experiments at elevated temperature (around 2000K) and pressure (35-100 GPa) in diamond anvil cells starting from mixtures of rhenium with different element and compounds ($\text{Re}+\text{N}_2$, NaN_3 , NH_4N_3) as well as characterizing the end products by single and powder diffraction, temperature-dependent electrical resistivity measurements, and nano-hardness measurements. In the latter case, the authors developed a synthesis protocol to produce the rhenium nitride pernitride crystals in larger sizes following a synthesis at high temperature and pressure in a large volume press. From the measurements reported, it is concluded that the ultraincompressible and hard rhenium nitride pernitride can be synthesized under extreme conditions and retrieved at room conditions. From the electrical resistivity measurements done as a function of temperature, it is concluded that the compound is indeed metallic. Calculations have also indicated the stability of this unique ReN_2 compound in the form of $\text{Re}_2(\text{N}_2)_2$. The high hardness, a property unusual of a metallic compound, can be explained by the formation of N-N covalent bonding. Directional N-N bonding was shown from electron density maps obtained computationally. The article raises the question of the possibility to obtain an ultra-low compressibility and high hardness in a metallic, hence challenging common wisdom in materials science. This may open new avenues to other compounds with unique and unusual properties.

The results presented are substantial and convincing. It should be pointed out, however, that the single X-ray diffraction data recorded from the synthesized rhenium nitride pernitride at elevated temperature and pressure used to determine the crystalline structure, could have been used, in principle, to generate experimental electron density maps. These electron density maps would be invaluable to actually show that directional N-N bonding is present and can be observed in the novel compound. Calculating and reporting the experimental maps would support the authors’ calculations presented in the article to explain the measured the unusual high hardness found in the new compound. Including such results in the present report, and this is believed to be doable, would make the conclusions of the paper even stronger to the readers. It would complement very well all other results presented. It is strongly suggested to the authors to consider this addition to their article prior to acceptance of their submission.

Although the work presented is very challenging, the article and supplementary material provide a valid analysis and the required level of information to reproduce experiments by other groups.

In brief, the authors have reported the formation of a novel compound resulting from the interaction of rhenium and nitrogen at elevated temperature and pressure with an unusual crystal chemistry and physical properties, namely, rhenium nitride pernitride $\text{Re}_2(\text{N}_2)(\text{N}_2)$. Both experimental and computational results were presented. As the compound shows ultra-low compressibility and high hardness which is certainly very unusual given its metallic nature, the findings reported here should be of great interest to the materials science community. Furthermore, the synthesis techniques developed and presented in this reported are certainly applicable to the preparation of other nitride and, as such, are of significant importance.

We thank all Reviewers for their valuable comments that helped to improve our manuscript. Below we provide a point-by-point reply to all raised questions. The modified parts of the text in the revised manuscript are highlighted in blue.

Reviewer #1

The manuscript by Bykov, et al. describes the preparation and characterization of $\text{Re}_2(\text{N}_2)(\text{N})_2$. This compound was first prepared on a small scale in a diamond anvil cell from rhenium metal and nitrogen gas. To prepare a sample on a bulk scale for mechanical properties characterization, the authors needed a new nitrogen source/precursor, NH_4N_3 , which they prepared from a metathesis reaction between sodium azide and ammonium nitrate. The ammonium azide was then combined with rhenium metal in a multi-anvil hot press. The resulting polycrystalline sample was then characterized via X-ray diffraction, and mechanical properties via nanoindentation (to yield a hardness of ~ 36 GPa) and compression under hydrostatic conditions (to yield a bulk modulus of 428 GPa). This paper should be accepted by virtue of their novel solid-state chemistry and mechanical properties, but the following minor points need to be clarified prior to publication.

Reply:

We thank the Reviewer for the positive evaluation of our work.

1) Could the work here be generalized to other late transition metals? E.g. could tungsten nitrides be analogously prepared with the same NH_4N_3 precursor?

Reply:

Sure, the NH_4N_3 precursor can be used not only for the synthesis of rhenium nitrides. Virtually all transition metals may react with nitrogen, which is released during decomposition of ammonium azide. It is well known that late transition metals like Os, Ir, Pt react with nitrogen above ~ 40 GPa yielding pernitrides OsN_2 , IrN_2 and PtN_2 . Although no experimental high-pressure studies at similar pressures have been performed for Hf, Ta and W, these metals are known to be readily able to form nitrides even at lower pressures. We mention this in the conclusions of the revised manuscript (page 10).

2) On page 6, line 138, the authors argue that there is increased ionicity in Re-N bonding, and this may be why ReN_2 has a higher bulk modulus when compared to OsN_2 , PtN_2 , and TiN_2 . I do not think ionicity is the appropriate descriptor as ionic character is usually a reflection of the difference in electronegativity, and one would expect that TiN_2 would be more ionic than ReN_2 as titanium (1.54) is less electronegative / more electropositive than rhenium (1.9). To reflect the unequal distribution of electron density, the authors should consider using the descriptor “polarization” instead of ionicity. Ionicity also conflicts with the author’s localization functions (see point #4 below).

Reply:

We do agree with the Reviewer. We indeed used an inappropriate term “iconicity”, but assumed the difference between the *formal* charges (oxidation states) of a cation and an anion. We modified the text on page 6 accordingly.

3) On page 6, lines 142-143, the authors argue that the increase in oxidation state results in an increase in the bulk modulus, as shown in their previous work on FeN_x. However, that is a bit counterintuitive, as lower oxidation states should have more electron density around the metal center, and thus should have a higher bulk modulus, not the other way around. From the author's previous work on FeN_x, one can rationalize the increase in bulk modulus comes from more electron density from more nitrogen, rather than an increase in oxidation state. Can the authors clarify their argument?

Reply:

We thank the Reviewer for this comment and would like to clarify the point to avoid a confusion. In our previous paper on iron nitrides (Bykov et al. Nat. Commun 9, 2756 (2018)) we indeed wrote: "Since, the compression of dinitrides is primarily controlled by the compression of metal-nitrogen (M-N) bonds^{34,35,37}, the dinitrides with weaker M-N bonds are expected to be more compressible... Therefore, the compressibility of M-N bonds should decrease in the following sequence: $M^{2+}-N > M^{3+}-N > M^{4+}-N$ ". Formal charges (oxidation states) were meant there, similarly to what we say in the present paper: larger formal charges lead to increase of the bulk moduli. In Bykov et al. (2018) we used this empirical relationship to judge the oxidation state of a metal when the compressibility of a nitride was known. Considering high formal charges of Re⁵⁺ and pernitride units [N-N]⁴⁻, the extremely high bulk modulus of ReN₂ agrees with the same trend, which provides a simple rule of a thumb for evaluation of the elastic properties of dinitrides just by looking at the Periodic table. The trend has been confirmed in numerous experimental and theoretical studies. In the revised manuscript we have added one more reference to Hazen, Downs and Prewitt (Reviews in Mineralogy and Geochemistry (2000) 41 (1): 1-33), who described this phenomenon (the larger the product of formal charges of cations and anions, the higher the polyhedral bulk moduli) to be relevant for many groups of compounds (oxides, silicates *etc.*), and reformulated the statement on page 6 in accordance with the discussion above.

4) On page 9, line 229, the authors argue that the "formation of strong covalent bond between Re and N₂ atoms," and this is confirmed with the author's electron localization functions. I agree with this assessment, and this leads me to think that the bonding is not ionic, but rather polar covalent.

Reply:

We agree with this comment of the Reviewer.

5) Can the authors comment on the stability of ReN₂ versus water and air in their main text?

Reply:

We have added the following sentence to the main text on Page 4:

After a stepwise decompression of the sample obtained in Experiment #1 down to the ambient pressure, all of the three phases (ReN₂, ReN_{0.6}, Re₂N) were found to remain intact even after being exposed to atmospheric oxygen and moisture for several months (Supplementary Figure 4)

6) Can the authors comment on how the measured density of their polycrystalline compacts compares to the theoretical crystal density in their main text?

Reply:

Unfortunately, the size of the sample is still too small to reliably measure the density of the polycrystalline compact.

7) The shape and size of the grains are extrinsic effects that can affect the nano-indentation hardness. Do the authors see any coarsening or texturing of the polycrystalline grains? The authors should include an SEM image of their sample. The synthesis of ReN_2 from ammonium azide should release hydrogen gas, do the authors see any porosity from trapped gasses?

Reply:

We have included a SEM image of the sample used for the nanoindentation measurements and a SEM image of the samples synthesized in a diamond anvil cell (Supplementary Figure 10). We do not see any porosity of the sample. We cannot recognize individual grains on the images and cannot comment on their shape.

Reviewer #2

Report on paper High-pressure synthesis of 1 ultraincompressible hard rhenium nitride pernitride $\text{Re}_2(\text{N}_2)(\text{N})_2$ stable at ambient conditions by Maxim Bykov *et al.*

The paper describes the high-pressure high-temperature synthesis of a new compound ReN_2 obtained in the search for new hard materials. Its bulk modulus K_0 of 428(10) GPa is at the threshold of superhard materials. Authors determined the structure of ReN_2 , characterized its properties in details and scaled up the process of its synthesis from the DAC to a large volume press. Authors also discuss the role of structural features in the nitrogen-rich compounds. This is a thorough, quite complex and demanding study and it adds a new representative to the group of metal nitrides and to the search for superhard materials. I have checked the crystallographic data and they are consistent.

Reply:

We thank the Reviewer for pointing out the thoroughness and challenges of our work.

On the other hand, this continuous search has been continued for some time, as apparent from the whole series of MN_x compounds, and one may ask, what is the real progress in the chemistry and practical aspects of the production of new superhard materials. ReN_2 is not the best, cheapest or technologically easiest to obtain material and for the practical and technological reasons this compound is not particularly exciting. It is interesting for the structural reasons, as a new member of the series. However I doubt if it would attract strong attention expected for Nature Comm articles. The paper contains lots of data measured and

calculated by several methods, but it is more a report of these properties than sublimed analysis explaining general science.

Reply:

ReN₂ is not only the most incompressible among metal pernitrides, but it also has a bulk modulus that is very close to that of diamond. Given its metallic properties and very high hardness, we are convinced that this is truly an exceptional material. Furthermore, the methodology of the synthesis can be definitely applied to the synthesis of dinitrides of other transition metals (see also our reply to the comment 1) of the Referee #1). We are happy that Referees #1 and #3 share our opinion.

The style of the paper could be improved, too; there are also some mistakes (there are no crystallographic sites 4e, but Wyckoff sites e, or general sites in this case, and 4 is the number of positions) and typos, like in line 61/page 2: binary systems M-N systems ...

Reply:

We are grateful to the Reviewer for pointing out the typos. They are corrected in the revised version of the manuscript.

Reviewer #3:

This report by Bykov *et al.* titled “High-pressure synthesis of ultraincompressible hard rhenium nitride pernitride $\text{Re}_2(\text{N}_2)(\text{N})_2$ stable at ambient conditions” provides evidence, both experimental and computational, for the formation of a new form of rhenium nitride, namely, $\text{Re}_2(\text{N}_2)(\text{N})_2$. This nitride pernitride compound is shown to be presenting ultra-low compressibility and high hardness. This discovery is unusual as it challenges the current understanding that metallic binding does not translate into high hardness in a MX_2 compound (where “M” represents a metallic element). In order to document their findings, the authors have carried out numerous rhenium nitride synthesis experiments at elevated temperature (around 2000K) and pressure (35-100 GPa) in diamond anvil cells starting from mixtures of rhenium with different element and compounds ($\text{Re}+\text{N}_2$, NaN_3 , NH_4N_3) as well as characterizing the end products by single and powder diffraction, temperature-dependent electrical resistivity measurements, and nano-hardness measurements. In the latter case, the authors developed a synthesis protocol to produce the rhenium nitride pernitride crystals in larger sizes following a synthesis at high temperature and pressure in a large volume press. From the measurements reported, it is concluded that the ultraincompressible and hard rhenium nitride pernitride can be synthesized under extreme conditions and retrieved at room conditions. From the electrical resistivity measurements done as a function of temperature, it is concluded that the compound is indeed metallic. Calculations have also indicated the stability of this unique ReN_2 compound in the form of $\text{Re}_2(\text{N}_2)(\text{N})_2$. The high hardness, a property unusual of a metallic compound, can be explained by the formation of N-N covalent bonding.

Directional N-N bonding was shown from electron density maps obtained computationally. The article raises the question of the possibility to obtain an ultra-low compressibility and high hardness in a metallic, hence challenging common wisdom in materials science. This may open new avenues to other compounds with unique and unusual properties.

The results presented are substantial and convincing. It should be pointed out, however, that the single X-ray diffraction data recorded from the synthesized rhenium nitride pernitride at elevated temperature and pressure used to determine the crystalline structure, could have been used, in principle, to generate experimental electron density maps. These electron density maps would be invaluable to actually show that directional N-N bonding is present and can be observed in the novel compound. Calculating and reporting the experimental maps would support the authors’ calculations presented in the article to explain the measured the unusual high hardness found in the new compound. Including such results in the present report, and this is believed to be doable, would make the conclusions of the paper even stronger to the readers. It would complement very well all other results presented. It is strongly suggested to the authors to consider this addition to their article prior to acceptance of their submission.

Although the work presented is very challenging, the article and supplementary material provide a valid analysis and the required level of information to reproduce experiments by other groups. In brief, the authors have reported the formation of a novel compound resulting from the interaction of rhenium and nitrogen at elevated temperature and pressure with an unusual crystal chemistry and physical properties, namely, rhenium nitride pernitride $\text{Re}_2(\text{N}_2)(\text{N})_2$. Both experimental and computational results were presented. As the compound shows ultra-low compressibility and high hardness which is certainly very unusual given its metallic nature, the findings reported here should be of great interest to the materials science

community. Furthermore, the synthesis techniques developed and presented in this reported are certainly applicable to the preparation of other nitride and, as such, are of significant importance.

Reply:

We thank the Reviewer for the positive evaluation of our manuscript.

Following the Reviewer's suggestion, in the revised version of the manuscript we provide an *.fcf* file generated with SHELX instruction LIST 3, in case the readers would like to visualize different sections of the electron density maps.

Still, we would like to be careful and not to over interpret these experimental data. While our X-ray diffraction data allows to locate both heavy Re (75 electrons) and light nitrogen (7 electrons) atoms, the analysis of the electron densities or residual density maps in order to make any conclusions about the chemical bonding is not justified. Inevitable residual electron density close to heavy atoms arise due to Fourier truncation errors. The Figures below demonstrate this.

Figure 1. Fourier map (left) and difference Fourier map (right) for ReN_2 at ambient conditions. The section is drawn through N1-N1-N2 atoms (plane (-1 3.87 1.51) with distance from origin 6.26 Å).

In this view, we are fully convinced that calculated electron density maps provide more reliable information about the bonding in the $\text{Re}_2(\text{N}_2)(\text{N})_2$ compound.

With best regards

Dr. Maxim Bykov

REVIEWERS' COMMENTS:

Reviewer #1 (Remarks to the Author):

The changes make this paper acceptable for publication.

Reviewer #3 (Remarks to the Author):

Here is my short report of the manuscript by Bykov et al, titled "High-pressure synthesis of ultraincompressible hard rhenium nitride pernitride $\text{Re}_2(\text{N}_2)(\text{N})_2$ stable at ambient conditions". I am reviewing here the corrected manuscript resubmitted following the initial evaluation from three referees. The present assessment of this resubmission is based on the referees' comments and objections and the reply and corrections presented the authors.

To my opinion, the authors have addressed satisfactorily the major points raised by all the referees. The manuscript has been corrected accordingly, in the text with the addition of figures to support the reply.

Overall, I am satisfied with the corrections. I do recommend the publication of the revised manuscript in Nature Communications.

REVIEWERS' COMMENTS:

Reviewer #1 (Remarks to the Author):

The changes make this paper acceptable for publication.

Reviewer #3 (Remarks to the Author):

Here is my short report of the manuscript by Bykov et al, titled "High-pressure synthesis of ultraincompressible hard rhenium nitride pernitride $\text{Re}_2(\text{N}_2)(\text{N})_2$ stable at ambient conditions". I am reviewing here the corrected manuscript resubmitted following the initial evaluation from three referees. The present assessment of this resubmission is based on the referees' comments and objections and the reply and corrections presented the authors. To my opinion, the authors have addressed satisfactorily the major points raised by all the referees. The manuscript has been corrected accordingly, in the text with the addition of figures to support the reply.

Overall, I am satisfied with the corrections. I do recommend the publication of the revised manuscript in Nature Communications.

Reply:

We thank the Reviewers for positive evaluation of our work